# An Environment-Specific Prioritization Model for Information-Security Vulnerabilities Based on Risk Factor Analysis

Jorge Reyes [1,*] , Walter Fuertes [1] , Paco Arévalo [2] and Mayra Macas [1,3]

1   Department of Computer Sciences, Universidad de las Fuerzas Armadas ESPE, Av. General Rumiñahui S/N, Sangolqui P.O. Box 17-15-231B, Ecuador; wmfuertes@espe.edu.ec (W.F.); mayramacas@ieee.org (M.M.)
2   Department of Mathematics, Universidad Tecnológica Equinoccial, Rumipamba y Bourgeois, Quito 170147, Ecuador; paco.arevalo@ute.edu.ec
3   College of Computer Science and Technology, Zhejiang University, No. 38 Zheda Road, Hangzhou 310027, China
*   Correspondence: jlreyes5@espe.edu.ec

**Abstract:** Vulnerabilities represent a constant and growing risk for organizations. Their successful exploitation compromises the integrity and availability of systems. The use of specialized tools facilitates the vulnerability monitoring and scanning process. However, the large amount of information transmitted over the network makes it difficult to prioritize the identified vulnerabilities based on their severity and impact. This research aims to design and implement a prioritization model for detecting vulnerabilities based on their network environment variables and characteristics. A mathematical prioritization model was developed, which allows for calculating the risk factor using the phases of collection, analysis, and extraction of knowledge from the open information sources of the OSINT framework. The input data were obtained through the Shodan REST API. Then, the mathematical model was applied to the relevant information on vulnerabilities and their environment to quantify and calculate the risk factor. Additionally, a software prototype was designed and implemented that automates the prioritization process through a Client–Server architecture incorporating data extraction, correlation, and calculation modules. The results show that prioritization of vulnerabilities was achieved with the information available to the attacker, which allows evaluating the overexposure of information from organizations. Finally, we concluded that Shodan has relevant variables that assess and quantify the overexposure of an organization's data. In addition, we determined that the Common Vulnerability Scoring System (CVSS) is not sufficient to prioritize software vulnerabilities since the environments where they reside have different characteristics.

**Keywords:** prioritization model; probability theory; risk factor; Shodan; vulnerability scanning; vulnerability detection

## 1. Introduction

The significant technological advance and the constant use of applications on the network increase the number of vulnerabilities that cybercriminals exploit daily. Fixing vulnerabilities requires a lot of effort, time, and resources [1,2]. The cybersecurity analysts in the CERT/CSIRT of the different organizations have an arduous task at the level of proactive services whose main objective is to prevent attacks before they happen [3]. Those responsible for security must also analyze what vulnerabilities affect IT assets. In this process, they generally face an overwhelming volume of openness, which represents a high complexity when they have several assets connected to different networks. Resource limitations prevent mitigating all but a small number of vulnerabilities in an enterprise network [4–6].

A wide variety of tools help in the vulnerability scanning and detection process [7,8]. Most of the results from these tools are the Common Vulnerabilities and Exposures (CVE)

records, which are stored in the National Vulnerability Database (NVD) [7]. NVD includes specific parameters such as solution information, severity scores via CVSS, and impact scores [4]. The CVSS Score is the global standard for characterizing and measuring the severity of security vulnerabilities. However, the efficiency of this metric is affected by additional environment variables present in computer networks. Thus, by itself, it is not a good predictor of vulnerability exploitation and probability of occurrence [1–3]. Additionally, due to the large number of vulnerabilities that NVD contains and the amount of information for each exposure, it is essential to maintain an analysis with as many variables as possible [9].

Vulnerability treatment is a critical process in network and software security management [10]. The key to success is prioritizing [3], since it speeds up attention to the vulnerabilities with the most significant impact on assets, optimizing resources and improving security [5]. The prioritization of vulnerabilities is a complex process, where the order of attention must be defined to remedy and minimize the risk. Various organizations, companies, and researchers have their own rating systems to classify and prioritize vulnerabilities based on qualitative and quantitative rating systems. Qualitative rating systems are an intuitive approach for describing the severity of vulnerabilities. On the other hand, quantitative rating systems associate a score with each vulnerability [11,12].

To address the prioritization problem focused on the organization and its environment variables which define a specific risk factor for the IT asset where the vulnerability is located, this study proposes the development of a prioritization model that uses Shodan as a vulnerability scanning tool. The input data are collected through its REST API due to the large amount of information that the tool provides. Therefore, it is necessary to carry out a process of extracting information related only to vulnerabilities. For the prioritization process, formulas are proposed that quantify the environment variables to obtain the probability of occurrence. More precisely, the variables proposed from the collected data are the following:

- Total number of vulnerabilities identified;
- Average organizational risk;
- Average remediation time;
- Number of vulnerabilities per IP;
- Open ports of the organization;
- Ports opened by IP;
- Query Tags by IP that Shodan crawlers have identified;
- Total number of references;
- Probability of exploiting a vulnerability.

To achieve a quantitative qualification, values must be assigned to the qualitative variables; in this way, the order of attention can be prioritized. Finally, a starting point is achieved for the review and remediation of vulnerabilities based on the probability of occurrence of the vulnerabilities in each network IP, accounting for the information available to any attacker.

Among the main contributions of this study, we can mention:

(i) The creation of a mathematical prioritization model that allows for assigning a risk factor to each vulnerability identified within a set of IPs for the same organization.

(ii) The design and implementation of the proposed model following a client/server architecture based on variables of the network environment and variables extracted from the set of identified vulnerabilities. This prototype has three modules that include data extraction, correlation, and prioritization algorithms.

The remainder of this article is organized as follows: in Section 2, related work is briefly overviewed. Section 3 explains the methodology for the prioritization process. Section 4 presents the methods and relevant aspects in developing the prototype that automates the prioritization process. In Section 5, the results achieved during the prototype execution and

the main findings are discussed. Finally, Section 6 ends with the conclusions and outlines future work.

## 2. State of the Art of Current Vulnerability Prioritization

The global standard used by organizations worldwide is CVSS, which has constantly evolved and improved. However, its constant development and research is proof of the difficulty of standardizing risk and impact in a way that can be applied to all organizations. CVSS consists of three groups of metrics: base, temporal, and environmental [13]. The first works in this area tried to leverage the characteristics of the vulnerability to determine the probability of exploitation [14]. Dondo [5] presented a fuzzy system approach based on vulnerability attributes that help assess the relative potential risk of computer network assets. The CVEs obtained through vulnerability databases known as NVDs are used as input data. Furthermore, fuzzy rules are employed to infer risk exposure and attack probability; this allows vulnerabilities to be classified and the priority of attention to be defined. He also mentions that the prioritization approach demonstrates more significant severity values than those calculated by the CVSS.

One of the fundamental objectives of prioritization models is to create automated processes that obtain high effectiveness percentages. Amankwah et al. [1,15] propose a new automated framework for assessing the severity of vulnerabilities in open-source web scanners. This study mentions that CVSS is criticized for its high sensitivity but low specificity for the exploits used and, therefore, the inconsistency in the severity score. For this reason, they propose four evaluation references metrics: impact, exploitability, prevalence, and detectability. The framework of this study has three components: vulnerability detection, vulnerability assessment based on the references mentioned above, and, finally, vulnerability prioritization according to the determined severity. The results show a more centralized approach achieving greater risk precision that allows prioritizing attention to vulnerabilities.

Sharma and Singh [12] propose a hybrid approach derived from the combination of Vulnerability rating and scoring system (VRSS) and CVSS with two temporary metrics: remediation level and vulnerability index. The input data are collected from NVD and consider the level of remediation and the rate at which a particular vulnerability grows over time, called the vulnerability index. The vulnerability index raises the static score to account for the rate of change in exposure over time. A quantitative score is obtained from qualitative variables for prioritization in this study.

Spanos et al. [9] propose a text mining process, which analyzes the description issued by NVD regarding known vulnerabilities. They apply three classification methods: decision trees, neural networks, and support vector machines. This study concluded that the description itself is a very accurate and reliable source of information for prioritizing vulnerabilities. Sharma et al. [11] also perform a similar process using a convolution neural network (CNN). Specifically, the authors try to prioritize the vulnerabilities based on keywords emitted in the description.

Deb and Roy [16] present a mathematical formulation through a Bayesian network in the software-defined networking (SDN) environment to identify the status of the different hosts on the network. The CVSS and the Bayesian network are robust methodologies for determining the mutual relationship between vulnerabilities and prioritizing the effective care process. On the other hand, Hu et al. [17] propose two algorithms, the threat prediction algorithm based on a dynamic Bayesian graph and the security risk quantification algorithm based on threat prediction. The first algorithm aims to provide comprehensive predictive information under a specific threat scenario. The second algorithm quantifies the threat in the first algorithm based on the security risk at two levels: host and network. The framework proposed in his study contains three components: situational factors collection, threat scenario prediction, and security risk situation quantification.

Chen et al. [18] mention that the average time it takes for NIST to issue a result on the risk of discovered vulnerabilities is around 132.7 days. To mitigate this long delay,

they propose a Vulnerability Analysis and Scoring Engine (VASE) system. The system is based on the results issued in cybersecurity forums on Twitter to collect the possible scores and average them, yielding tentative results about the vulnerability rating in question. VASE adopts the convolutional graph network (CGN) model, where the nodes correspond to CVEs. VASE consists of three main components: graph construction based on basic natural language processing methods, attention-based input embedding, and transductive inference with GCN-AE.

It is essential to note that the input data are massive for several studies described above. That is, the applied models focus on all vulnerabilities and their characteristics. Our study tries to contribute additional evaluation values to the CVSS model, making it more specific when defining risk and impact. Moreover, the studies analyzed so far lack an approach that addresses organizational variables. CVSS remains a generic model that does not address the specificity necessary for successful prioritization in organizations [1–3].

Aiming to address the aforementioned problem, Farris et al. [10] present a software called VULCON, which has a strategy based on two fundamental performance metrics: time-to-vulnerability remediation (TVR) and total vulnerability exposure (TVE). The authors use a mixed-integer multi-objective optimization algorithm to prioritize vulnerabilities. Their study analyzes the environment variables where vulnerabilities are found and detected. The different variables contained in the vulnerabilities, such as the year shown in the CVE-ID, are essential when prioritizing. In addition, the ports corresponding to the network's IPs are considered since there is a greater probability of use and exploitation. VULCON tries to add environment variables that improve the prioritization quality each time. The achieved results demonstrate higher quality in prioritization with the additional environment variables.

One of the fundamental factors when forming a correct prioritization is the quality of the information. The more specific it is, the greater the likelihood of successful prioritization. The objective is to propose generic vulnerability prioritization models that evaluate the organization, leveraging and relying on the information provided by the collected data [11,12]. This approach means that prioritization is not biased in any direction, providing a justification panorama according to which cybersecurity experts in organizations do not address vulnerabilities that represent a "lower risk". Finally, the data identified in the scanning processes will only predict an organization's actual risk.

Compared to our study, the analyzed works provide relevant information about certain network environment variables and how they are quantified [10,18]. They also allow the definition of validation rules based on the results obtained in [5,14,17]. On the other hand, one of the approaches is to propose improvements in the CVSS model [1–3]. However, keeping CVSS as a variable has been identified to help optimize prioritization time as it extensively describes the impact of vulnerability [10–12]. Some studies focus on proposing prioritization models that have as input a set of specific test vulnerabilities extracted from NVD [1,9,12,16]. Unlike these works, our study focused on calculating the risk factor to prioritize vulnerabilities identified for any organization by understanding the environmental variables included in the data. For this reason, the order of prioritization that we propose is generic and easily adaptable to changes in the continuous vulnerability detection process since it can correlate the data of the same organization.

## 3. Research Methodology

We employed the OSINT framework for data collection, analysis, and knowledge extraction. This search process aims to prioritize the vulnerabilities exposed to any user on the Internet, which can be collected openly. Similarly, we use the Shodan search engine to obtain relevant information about existing vulnerabilities of different IPs of the computers within the same domain or organization. Due to the large amount of data that a query can return through Shodan's REST APIs, we have applied specific algorithms that allow us to extract information regarding only the vulnerabilities and characteristics of the environment where they have been identified. Later, we apply mathematical formulas to

quantify the qualitative variables. Once the environment variables have been mapped to the vulnerabilities, attention can be processed and prioritized. In this way, an added value is included in the management of vulnerabilities, taking into account that the actions of cyber criminals begin with the data and information collected openly. A brief description of the methodological process followed is presented below.

### 3.1. OSINT Framework

Open Source Intelligence (OSINT) is a framework that allows collecting, processing, and correlating information from open sources from all over cyberspace to generate knowledge. Technological advances make OSINT evolve at a dizzying pace, providing innovative applications driven by data and Artificial Intelligence for different areas such as politics, the economy, or society. This framework also offers new lines of action against cyber threats and cybercrime [19]. OSINT has three representative phases that define the information processing methodology. Figure 1 describes these phases.

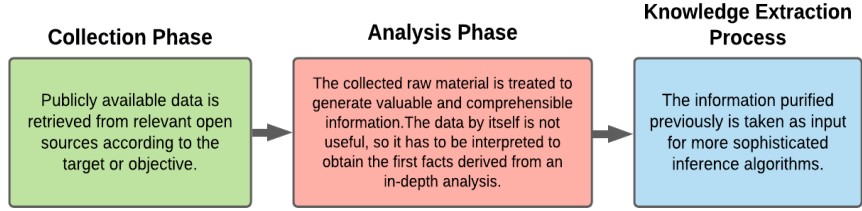

**Figure 1.** OSINT Phases [19,20], sequential phases for open-source information processing. The objective of this process is to generate knowledge from the data collected.

### 3.2. Collection Phase

We use Shodan, a cloud-based computer security scanner for the collection phase. Shodan is a security software with several search algorithms and relies on various Open Source tools that provide extensive information about the hosts detected by its crawlers [20]. It is regarded as the world's first search engine for Internet-connected devices [21]. In Shodan, all one has to do is enter the org code: "Organization _Name" and a series of IP addresses begin to be collected, from which information will be obtained.

#### 3.2.1. Shodan

Shodan provides valuable information to cybersecurity researchers in organizations, but also to users with malicious intentions since, being a search engine, it is available to the general community [22]. It has a database that stores all the information collected by its crawlers about the different IPs that they track on the Internet. Due to the information collected from Shodan, many experts choose to use techniques that hide the different IPs of the crawlers; in this way, they avoid being detected. On the other hand, some experts use this valuable information to apply corrective measures that improve the security of their networks.

Shodan provides a REST API to make general or specific queries according to the needs of the developers. It returns a JSON response that facilitates the manipulation and extraction of data. In addition, it has an automated notification process, which alerts and notifies the different results that it monitors, complying with an early detection service. Shodan can display 11 general variables via a banner provided by the REST API, depending on the scanned IP and its detection. These variables, in turn, contain other properties, resulting in more than 50 variables in total. Table 1 lists the 11 general variables that are relevant to this analysis process; this information is directly related to the knowledge and understanding of vulnerabilities and the environment where they have been identified.

**Table 1.** Variables of Shodan Banner Specification.

| Variable | Description |
|---|---|
| domains | An array of strings containing the top-level domains for the hostnames of the device. |
| hostnames | An array of strings containing all of the hostnames that have been assigned to the IP address for this device. |
| org | The name of the organization that is assigned the IP space for this device. |
| data | Contains the banner information for the service. |
| city | The name of the city where the device is located. |
| isp | The ISP that is providing the organization with the IP space for this device. Consider this the "parent" of the organization in terms of IP ownership. |
| last_update | Date and time of the last IP update/revision |
| vulns | An array of strings containing the CVE code of the detected IP vulnerabilities. |
| country_name | The name of the country where the device is located. |
| ip_str | The IP address of the host as a string. |
| ports | The port number that the service is operating on. |

During this phase, an attempt is made to obtain the details of each vulnerability identified. The variables that a vulnerability contains are described below:

Common Vulnerabilities and Exposures (CVE)

CVE is a list of records released by MITER Corporation in 1999. Each record has an identification number, a description, and at least one public reference for publicly known vulnerabilities. CVE records are used in numerous cybersecurity products and services around the world. The CVE ID syntax is made up of $CVE + year + sequence\ number$. It is important to mention that the year does not indicate when the vulnerability was discovered but only when it was made public or assigned. At the same time, the number sequence is the unique identifier by year. Additionally, CVE includes a unique description, which generally contains details such as the name of the affected product and vendor, a summary of the affected versions, the type of vulnerability, the impact, the access required by an attacker to exploit the vulnerability, and the code components or important inputs that are involved [23]. In Table 2, we present a representative CVE comprised of two fields, CVE ID and description.

**Table 2.** Example of CVE.

| Field | Value |
|---|---|
| CVE ID | CVE-2017-9798 |
| CVE Description | Apache HTTPD allows remote attackers to read secret data from process memory if the Limit directive can be set in a user's .htaccess file, or if httpd.conf has certain misconfigurations, aka Optionsbleed. This affects the Apache HTTP Server through 2.2.34 and 2.4.x through 2.4.27. The attacker sends an unauthenticated OPTIONS HTTP request when attempting to read secret data. This is a use-after-free issue and thus secret data are not always sent, and the specific data depend on many factors including configuration. Exploitation with .htaccess can be blocked with a patch to the ap\_limit\_section function in server/core.c. |

Common Vulnerability Scoring System (CVSS)

Each CVE is assigned a value by a scoring system designed to provide an open and standard method for estimating the impact derived from vulnerabilities. It also helps

to quantify the severity that vulnerabilities may represent. This scoring system obtains standardized vulnerability values for CVEs ranging from 0 to 10, with 0 being the lowest and 10 considered critical. In this way, consistent criteria can be maintained for managing weaknesses in hardware and software that have been evaluated.

The CVSS score is calculated by combining several vulnerability characteristics called CVSS metrics [24]. By using an open framework, it is possible to know the characteristics of each vulnerability. However, being a general overview, there are no variables specific to the environment where they have been identified. A method of this nature contributes to having a broad picture of an organization's exposure to risks, which can arise from vulnerabilities that have already been identified and assessed.

*3.3. Analysis Phase*

An organization can have *i* vulnerabilities, with each vulnerability having *j* variables. In addition, the set of vulnerabilities can be stored as an IP with *k* variables, which allows collecting and defining information regarding the environment where they have been detected.

The collected dataset allows the understanding of the environment variables of the organization in question. For this study, prioritization variables are analyzed, extracted directly from the set of vulnerabilities and data detected in the collection phase. Next, the environment variables extracted from the collected data are presented.

3.3.1. Global Variables

The set of vulnerabilities found in the same network is an indicator of quantitative values to extract knowledge of the environment variables where they have been detected. Below are mathematical formulas that allow us to assign global values to vulnerabilities.

Total Vulnerabilities (TV)

Any public discussion of information about vulnerabilities can help a hacker. An organization uses many resources and works to protect its networks and fix all possible holes. It is easier for a hacker to find a single vulnerability, exploit it, and compromise the network. About 52% of exploited vulnerabilities are discovered by the direct action of the cybercriminal [25–28].

More and more highly sophisticated vulnerability exploit tools are being made public. For example, the National Security Agency (NSA) security tools leaked in 2016 contained hundreds of sophisticated exploits and back doors to vendor systems [29,30]. At the same time, patching or quickly updating vulnerabilities is impractical in domains such as critical infrastructure networks due to their high availability demands. In addition, an organization's risk increases when it has many unaddressed vulnerabilities. Considering that the global average cost to organizations for a data breach and vulnerability exploitation is USD 86 million [25,26], it can be determined that the number of vulnerabilities increases the risk of the organization because of the exploitation probability increases.

Shodan displays the number of vulnerabilities identified on each IP it tracks. For this reason, this variable is the growth or decrease rate of the general risk of an organization [12,23,31]. For this reason, in Equation (1), the following summation is presented:

$$tv = \sum_{i=1}^{N} v_i \tag{1}$$

where $N$ is the number of scanned IPs, and $v_i$ represents the number of vulnerabilities identified in the IP.

Average Organizational Risk (AOR)

The CVSS scoring system is widely researched and used in most organizations worldwide, representing the risk for each vulnerability [32,33]. However, as mentioned, its score

is focused on vulnerabilities. For this reason, organizations use their additional evaluation criteria to be able to define the actual impact on their environment [1–3].

Since CVSS has a quantitative scoring system, vulnerabilities can be grouped according to their criticality range as shown in Table 3. Using Equation (2), it is possible to obtain the score that demonstrates the organizational risk based on the vulnerabilities contained. In addition, an average value of the risk to which the organization is exposed can be obtained since all the identified vulnerabilities are immersed in the same environment.

**Table 3.** Correspondence between CVSS score and qualitative value (severity).

| Score | Severity |
|---|---|
| 0 | Null |
| 0.1–3.9 | Low |
| 4.0–6.9 | Medium |
| 7.0–8.9 | High |
| 9.0–10.0 | Critical |

More precisely, the average risk of the organization defined by CVSS is calculated as follows:

$$aor = \frac{\sum_{i=1}^{N} \sum_{j=1}^{v_i} (CVSS_{ij})}{tv} \tag{2}$$

where $N$ represents the number of IPs in the organization and $v_i$ is the number of vulnerabilities contained in each IP. As can be seen, it is necessary to add each $CVSS_{ij}$ that contains a vulnerability. The resulting $AOR$ value is mapped to the ranges presented in Table 3 to measure the severity of the organizational risk.

Average Vulnerability Time (AVT)

The remediation time reflects the extent to which an organization is prepared to deal with a vulnerability. The level of preparedness of an organization for a vulnerability can significantly affect the severity associated with the vulnerability. Therefore, it is an essential variable for prioritizing vulnerabilities [12]. It is important to note that generally there is no security patch available when the vulnerability is released. For this reason, the severity score of a vulnerability is adjusted downward, suggesting a decreasing level of urgency as the remediation becomes final. The less official the fix, the higher the vulnerability score [23]. However, this logic allows for defining that, if this repair is not applied in the organizations, the risk of that environment will remain adjusted to discharge because the initial impact scenario is maintained.

The average time to exploit an unpatched vulnerability in systems is rapidly decreasing. According to reports submitted by different industries, it takes about 15 days to exploit a vulnerability by cybercriminals since its discovery [34]. The organizations targeted by cybercriminals require a more significant effort to correct attack vectors. For this reason, it is not recommended to have a vulnerability exposed for a long time period. The average time it took for an organization to detect and remediate a vulnerability was 180 days in 2018, and, for 2020, it was 280 days, indicating that it is increasing [26,27]. On the other hand, the vulnerabilities in the respective CVE ID offer information regarding the year they were made public. This helps to identify the extent to which the organization is dealing with known vulnerabilities. Regardless of the year in which a specific technology has been implemented, the CVE-ID will demonstrate that said service or software has been vulnerable for some time; therefore, it is advisable to update it. If good security practices are ignored, the probability of being attacked increases [34]. Therefore, this time variable is

essential in the prioritization process. The average time of the vulnerabilities is calculated as follows:

$$avt = \frac{\sum_{i=1}^{N} \sum_{j=1}^{v_i} (Current_{\text{year}} - CVE_{\text{year}_{ij}}) * 365}{tv} \tag{3}$$

Since the resulting AVT value corresponds to an average in days, it must be adjusted to a range of values between $0 \leq avt \leq 1$. In this way, the probability can be evaluated, taking the average detection and remediation time for the organizations described above as a reference. Accordingly, the following conditions are proposed:

- $avt \geq 280 \rightarrow avt = 1$
- $140 < avt < 280 \rightarrow avt = 0.5$
- $avt \leq 140 \rightarrow avt = 0.1$

The average detection time with which the comparison is made is referential and will change over the years. It is vital to consider when executing the prioritization process since the average time is the one previously proposed for the ongoing process of this study.

### 3.3.2. IP Variables

These variables are generated from the information obtained only from the scanned IP and directly affect the set of vulnerabilities identified within the IP.

#### Probability of Occurrence of an Event in the IP (POE)

Given that each IP in the network is independent [10] and assuming that the probability of occurrence of an event is independent of the IP, we define Equation (4):

$$poe_{\text{ip}} = \frac{v_{\text{ip}}}{tv} \tag{4}$$

where $v_{\text{ip}}$ represents the number of vulnerabilities in the IP. As can be seen, the greater the number of vulnerabilities, the greater the probability of an event occurring, regardless of the impact it represents.

#### Probability of Open Ports (POP)

The protection of information and the high availability of services require excellent technical and technological effort. Losing or exposing information or leaving services inactive harms organizations, both at a functional and a reputational level [35]. The rapid digital transition exposes vulnerabilities that are being exploited by cybercriminals [36,37]. Common security incidents include malware infection, ransomware, exploit exploits, improper access to applications, social engineering attacks, and denial of services [38,39].

Shodan detects the different open ports that a given IP has. The ports represent information exchange and communication vectors; they identify the process to which messages within the machine should be delivered. For this reason, the ports that are open and exposed directly to the public on the Internet represent a greater risk and, therefore, a greater priority for attention and control since they allow the exchange of information. In order to obtain a quantitative value, the following equations are defined:

$$top = \sum_{i=1}^{N} op_i \tag{5}$$

where $op$ is the number of open ports in an IP; therefore, $top$ represents the total number of open ports in the organization with $N$ IPs.

Since each IP is independent in the network, the risk of open ports is defined for each IP as follows:

$$pop_{\text{ip}} = \frac{op_{\text{ip}}}{top} \tag{6}$$

Probability for Query Tags (PQT)

While collecting IP addresses of the target organization of this study, it was observed that Shodan returns a variable called "tags". This variable contains character strings that refer to the service found or hosted on the scanned IP. For the target IPs of this study, "tags" such as "database" and "self-signed" were demonstrated. This field was taken as a prioritization variable because, when a "tag" is detected in an IP, the Shodan variable "data" contain more specific information about the service—for example, versions, technologies, Operating Systems, and among other characteristics hosted on that IP.

Figure 2 illustrates the trend in the IPs presenting this variable. When the IPs have the "tag" variable, there are more data items. The number of vulnerabilities is greater because Shodan knows more specific organization data and can relate a more significant number of known vulnerabilities.

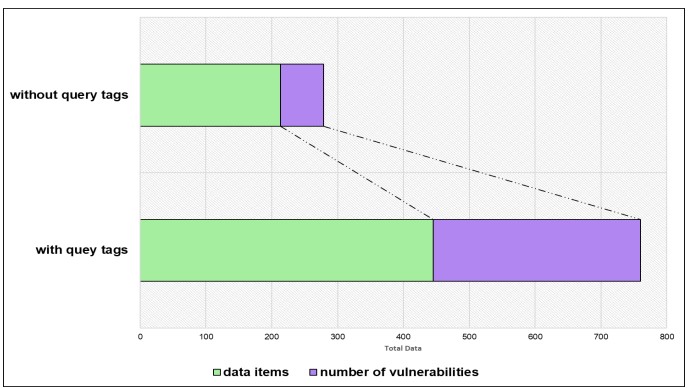

**Figure 2.** Number of data items and vulnerabilities when Query Tag is displayed in Shodan. The trend is directly proportional.

Because Shodan is a secure software and its documentation only describes the variable "tag" for the query processes and not in the response processes, we have limited ourselves to verify its existence in order to assign a risk value for the prioritization process [21,40]. Since the quality of information exposed in Shodan provides more significant value in knowledge and investigation for cybercriminals [20,22], the risk increases when this variable called "tags" has some content. In order to obtain quantitative values, the following conditions are proposed:

- $IF\ (tags > 0) \rightarrow pqt = 1$
- $IF\ (tags = 0) \rightarrow pqt = 0$

### 3.3.3. Vulnerability Variables

These types of variables are generated from the information of each vulnerability and affect only the vulnerability in question.

Total References (TR)

Each CVE record includes references where a broader context about the vulnerability can be understood. References should point to content relevant to the vulnerability and include at least all the details included in the CVE record. A key feature is that the references must also be publicly available [23].

There are two approaches to this variable. On the one hand, references are sources of valuable information when proceeding with the correction or mitigation of a vulnerability. On the other hand, this same information allows the attacker to know first-hand which products and versions are affected. In several cases, it has also been identified that the reference presents the exploits with which these vulnerabilities can be exploited [4,12].

For our case study, we employ the second approach mainly because, when applying OSINT techniques, we propose obtaining the most significant amount of information that

allows for compromising the organization's integrity, availability, and confidentiality. In addition, we maintain the attackers' point of view, where all this information is accessible without any limitation and allows them to carry out an intelligence process to understand the technological infrastructure. Based on the proposed approach and considering that the average number of references of a vulnerability with broad understanding is more significant than 8 [18], the following conditions are proposed:

- IF $(V_{\text{references}} \geq 10) \rightarrow tr = 1$
- IF $(V_{\text{references}} < 10) \rightarrow tr = \left( \frac{V_{\text{references}}}{10} \right)$

Exploitation Probability (EP)

For the identification and detection of vulnerabilities, researchers use exploits, which are codes that show the existence of a flaw; that is, confidentiality, integrity, or availability may be compromised. A CVSS score is indicative of the severity of the vulnerability but does not help predict the delay of the exploit [29].

According to Frei [41], while 94% of exploited vulnerabilities had an exploit available within 30 days, only 72% of patches were available within that period. Therefore, this is an indicator that there is an exploit available for the vast majority of old vulnerabilities. CVSS scores do not allow efficient discrimination between the probability of exploitation and non-exploitation [14,42]. However, studies have shown that high-risk vulnerabilities are more likely to be exploited [29,43]. In other studies, this variable is also known as the age of vulnerability [10]. Given that $CVE_{\text{year}} \leq Current_{\text{year}}$, the following conditions are proposed to prioritize attention to vulnerabilities:

- IF $(CVSS \geq 7.0) \rightarrow ep = 1$
- IF $(4 \leq CVSS \leq 6.9) \rightarrow ep = 0.5$
- IF $(0 \leq CVSS \leq 3.9) \rightarrow ep = 0.1$

3.3.4. Knowledge Extraction Process

Prioritization algorithms, models, and mathematical formulas cause many difficulties when applied to a real-world environment due to the organizations' multiple and conflicting objectives. More precisely, the different problems at the prioritization level have enormous practical implications since they produce a set of solutions based on the importance assigned to each of the business objectives [10].

According to the target organization, this study proposes a prioritization model based on the knowledge that the vulnerabilities themselves and the specific characteristics of the environment where they reside can offer us. Based on the set of vulnerabilities detected, the quantitative variables presented above are obtained, which contain hidden knowledge about the management and risk of the organization in question.

In order to quantify each vulnerability based on the identified environment variables, the risk factor that each one represents for the organization is calculated. The corresponding calculation process is described below:

Risk Factor (RF)

Two vectors define the risk factor, namely the probability of occurrence and the impact [2,44,45]. The following equation determines the *RF*:

$$RF = Probability\ of\ occurrence(po) \cdot Impact(i) \tag{7}$$

In this study, three types of variables are distinguished:

- Global Variables;
- IP Variables;
- Vulnerability Variables.

Determining the probability of occurrence by grouping the variables according to the type to which they belong is essential to keep the calculation focused on the level of impact,

i.e., whether it affects the vulnerability, IP, or network level. Considering that, for this case study, importance or superiority has not been defined by the type of variable, it is possible to average the values as presented in Equation (8):

Probability of occurrence:

$$po = \left( \frac{tr + ep}{2} \right) \cdot \left( \frac{poe + pop + pqt}{3} \right) \cdot avt \tag{8}$$

The impact is a widely researched and studied parameter in the CVSS framework, where various metrics are established to assess vulnerabilities and how these would affect the elements of the IT security environment if they materialize [13,46,47]. Considering that the objective of this study is to rely on the CVSS score to approximate a more exact value focused on the affected environment, the following equality $i = CVSS$ is determined.

Finally, the prioritization values assigned to the vulnerabilities after calculating the $RF$, are grouped according to a criticality range presented in Table 3. In Figure 3, the $RF$ is graphically presented by ranges.

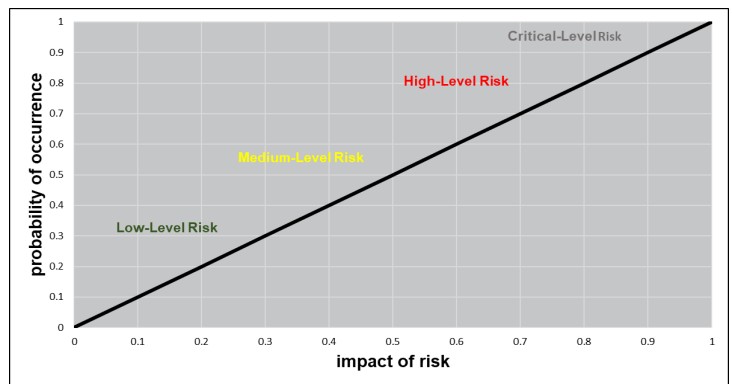

**Figure 3.** Risk Factor. The risk factor increases the criticality of a vulnerability when the probability of occurrence and impact is close to 1 [44].

Illustrative Example

After processing the information for our case study, the Risk Factor represents knowledge. When this process is applied to the total set of vulnerabilities, a quantitative prioritization is achieved that is close to the organizational reality due to incorporating environmental variables. For illustrative purposes, the calculation of a specific vulnerability is detailed. For this case, we will take the vulnerability CVE-2017-9798 determined in the previous section.

The organization that was the object of this study presented the following variables:

- Global Variables
    - $tv = 541$
    - $avt = 1$
- IP Variables
    - $poe = \left( \frac{96}{541} \right) = 0.18$
    - $pop = \left( \frac{9}{14} \right) = 0.64$
    - $pqt = 1$
- Vulnerability Variables
    - $tr = 1$
    - $ep = 0.5$
    - $CVSS = \frac{5}{10} = 0.5$

$$RF = \left( \frac{1+0.5}{2} \right) \cdot \left( \frac{0.18+0.64+1}{3} \right) \cdot 1 \cdot 0.5$$

$$RF = 0.23$$

Finally, taking the information available to the attacker as evaluation parameters, the *RF* of the previously presented vulnerability is less than the reference value that CVSS has assigned. This scenario will be discussed in detail in the Results section.

## 4. Prototype Development

The number of vulnerabilities detected in organizations is constantly increasing. The rapid digital transition organizations face forces staff to make hasty configurations and deployments, subject to continuous testing and change. These processes are constant, which explains why vulnerabilities appear and disappear with each phase of work.

The mitigation and treatment of vulnerabilities make the technological infrastructures improve or maintain their levels of security. The scanning, detection, and prioritization of vulnerabilities must be continuous and automated since the environment variables change according to the values obtained. Without a doubt, it is a great challenge for organizations to maintain these processes in such changing environments.

Previous studies have identified no methodologies or development processes defined for this type of system. However, the methodologies applied for the development process must show rapid results, constant changes, and evaluations during their construction. Due to being a complex data treatment process, it is essential to seek increasingly efficient results. On the other hand, from several studies analyzed, we conclude that 48% apply prototypes to evaluate the proposed processes because it reduces costs and time; the results are continuously evaluated and adjusted with the required changes. Figure 4 shows the phases proposed in our previous study, which we will follow for the development of the prototype.

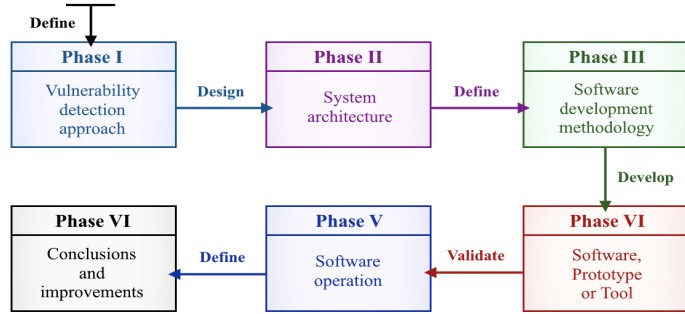

**Figure 4.** Development Process Phases. The six-phase process allows us to constantly evaluate the software, control complexity and risks, and increase its functionality throughout the process.

The screening approach for this study has been presented in detail in Section 3. However, it is crucial to consider that, according to the classification made in our previous study, we propose a mixed approach that takes advantage of information from known vulnerabilities and characteristics of information overexposure at the network level. Because Shodan provides extensive information and the approach for this study is mixed, we use its variables in a combined way to prioritize the order of attention of the vulnerabilities detected.

Shodan's documentation mentions that all their websites are entirely built on the same public API; this means that anything that can be done through the website can also be done programmatically using the API. Figure 5 shows a high-level diagram illustrating the flow of data on the Shodan platform.

According to the operating concept of Shodan's websites, we can see that Shodan itself is a server where the data found by its crawlers are stored. In this way, the data already present a preprocessing and mapping. However, it is still very scattered information, so the official website does not show all the API returned information.

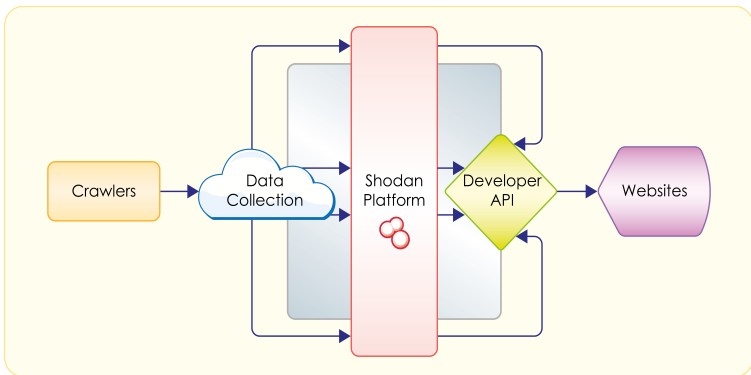

**Figure 5.** High-level diagram showing the flow of data in the Shodan platform [21]. Shodan's communication is unidirectional as it only allows client-side data queries.

In Figure 6, we propose a Client/Server architecture, where we make the necessary queries to Shodan to obtain the information corresponding to the IPs of the same organization. The queries start with a data extraction and analysis process, presented in a web service through three modules that show the correlated information about the detected vulnerabilities. The objective of these modules is to provide knowledge to the client about a specific organization. The correlation of vulnerabilities and network characteristics avoids the dispersion of data that have no relevance and require human capital to be interpreted and analyzed. It is possible to optimize resources, especially when knowing, interpreting, and prioritizing the vulnerabilities that Shodan has detected.

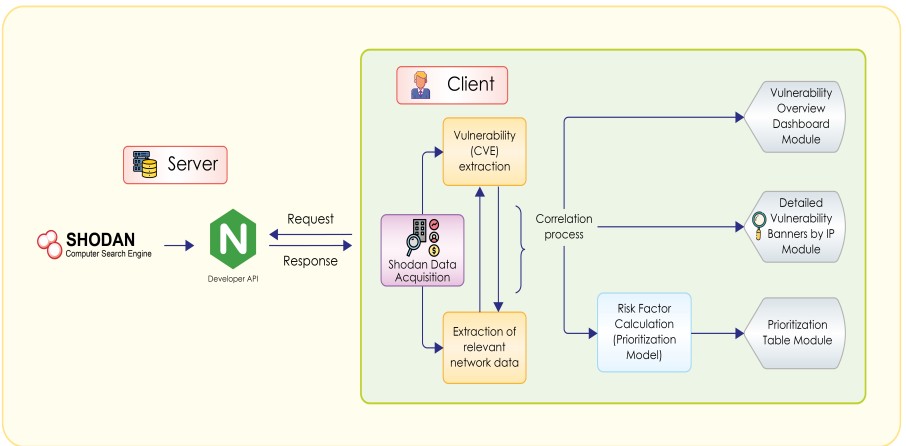

**Figure 6.** Architecture of the prototype's resource consumption and data processing diagram.

The prototype was developed in React, an open-source Javascript library designed to create user interfaces to facilitate the development of applications on a single page. Facebook and the free software community maintain it. We have published the source code of the individual modules and complete framework for further use by the respective community in a public repository (https://github.com/jorgereyesn/prioritization--model-shodan-jreyes-web-app.git (accessed on 2 December 2021). The ease of interacting with frontend and backend code allows dynamic testing quickly.

Shodan returns the information in JSON format; thus, a mapping process is necessary for its presentation and interpretation. In this study, we focus on the extraction of vulnerabilities following the architecture presented in Figure 6, which defines a process for extracting only the vulnerabilities found within all the collected data. To that end, Algorithm 1 was developed for extracting all the information related to the CVEs from the resulting JSON of Shodan.

---

**Algorithm 1:** Extract Vulnerabilities.

---

**Data:** Variable *data* and *vulns* of Shodan
**Result:** Vulnerabilities with description, CVSS and references (*returnData*)
$tempValue \leftarrow Array[empty]$;
$returnData \leftarrow Array[empty]$;
$count \leftarrow 0$;
**for** $i \leftarrow 0$ , *data.length* **do**
   **if** *data[i].vulns* $\neq$ *undefined* **then**
      | tempValue.push(data[i].vulns);
   **end**
**end**
**for** $i \leftarrow 0$ , *data.length* **do**
   $count \leftarrow 0$;
   **for** $j \leftarrow 0$ , *tempValue.length* **do**
      **if** *tempValue[j][vulns[i]]* $\neq$ *undefined* **then**
         **if** *vulns[i].index = tempValue[j][vulns[i]].index and count < 1* **then**
            tempValue[j][vulns[i]].cve $\leftarrow$ vulns[i];
            returnData.push(tempValue[j][vulns[i]]);
            count ++;
         **end**
      **end**
   **end**
**end**

---

Once the vulnerability extraction function has been defined, the function responsible for connecting to Shodan is called. Algorithm 2 is in charge of connecting, mapping, and extracting the information by making use of the algorithm mentioned above—Algorithm 1. The objective of this process is that, after Shodan returns the information to us, two values can be separated and returned that will contain only the necessary information on vulnerabilities and network, which facilitates the treatment and correlation for the prioritization and presentation processes on the screen.

---

**Algorithm 2:** Shodan Data Acquisition.

---

**Data:** *shodanData* in JSON format from
      https://api.shodan.io/shodan/host/*ip*?key=API_KEY where *ip* is input
      variable
**Result:** Network and Vulnerabilities values ($\{network, vuln\}$)
$vuln \leftarrow Array[empty]$;
$network \leftarrow Array[empty]$;
**if** *shodanData.vulns* $\neq$ *undefined* **then**
   network.push(shodanData);
   vuln.push(extractVulnerabilities(shodanData.data , shodanData.vulns));
**end**

---

Finally, the formulas presented in Section 3 were transformed into algorithms for the prioritization process, applying cyclical flow control structures to calculate the necessary values and assign them as variables to each vulnerability. Algorithm 3 is in charge of assigning the *RiskFactor* to each vulnerability to finally order them, achieving a prioritization based on the information that has been detected as available to the attacker.

---

**Algorithm 3:** Risk Factor Calculation

---

**Data:** $info = \{network, vuln\}$ with $tr, ep, poe, pop, pqt$ and $avt$ values
**Result:** $vulnerabilities$ with risk factor value ($info$)
**for** $i \leftarrow 0$ , *info.length* **do**
   **for** $j \leftarrow 0$ , *info.vuln.length* **do**
     info[i].vuln[j].po ← ((info[i].vuln[j].tr + info[i].vuln[j].ep)/2) *
     ((info[i].network.poe + info[i].network.pop + info[i].network.pqt)/3) * avt;
     info[i].vuln[j].impact ← info[i].vuln[j].cvss /10;
     info[i].vuln[j].rf ← info[i].vuln[j].po * info[i].vuln[j].impact;
   **end**
**end**

---

## 5. Results Analysis

This section describes the input data to check the operation of the prototype. In addition, the results obtained in the prioritization process are analyzed according to the proposed model.

### 5.1. Validate Software Operation

Considering that the prioritization model is based on the data available to the attacker, some variables related to the case study organization will not be shown for security reasons. This study proposes a generic model that can be applied to any organization. As such, in order to begin with detecting and analyzing vulnerabilities, it is enough to know a group of IP addresses that are part of the technological infrastructure.

To define the input data, it is necessary to carry out previous research in Shodan through its official website, where the following search string must be entered: $org$ : *"Organization_Name"*. Once the different IPs that are visible in Shodan have been collected, we enter them into the prototype to perform the necessary queries and map the information to be displayed in the modules. For this case study, nine IP addresses have been discovered, containing vulnerabilities and belonging to the same organization. As previously mentioned, the three component modules and the overall employed architecture are presented and described in Figure 6.

The *Vulnerability Overview Dashboard Module* shows general information related to the network and the detected vulnerabilities. This is achieved by applying Algorithm 1, which is responsible for extracting vulnerability and network information to be analyzed and mapped into fields. The objective of this module is to provide a global vision of the resulting data. Of the nine IPs scanned, 541 vulnerabilities have been detected, the same ones verified in Shodan directly through its website. The set of vulnerabilities shows an organizational risk of 5.29, which, according to Table 3, corresponds to medium severity. In addition, the average time of the identified vulnerabilities exceeds 2300 days.

After obtaining the general information about the set of detected vulnerabilities, it is important to have detailed information on how they were detected, i.e., the vulnerabilities for each specific IP. The *Detailed Vulnerability Banners by IP Module* contains in detail the information related to each scanned IP and the vulnerabilities that have been identified in it. Furthermore, the links of the vulnerabilities are referenced towards NVD for a more detailed investigation, the IP plus the ports are concatenated for a review of the content hosted in each port, and a link is generated according to the detected hostname, also for investigative purposes.

Finally, the *Prioritization Table Module* contains every vulnerability and the corresponding calculated prioritization variables. This is achieved using Algorithm 3 plus some simple calculations to obtain the values of the environment variables. The detected vulnerabilities have been ordered from highest to lowest (i.e., according to the risk factor evaluated) to suggest the user's order of review. In Table 4, an extract of the module found in the prototype is presented. It is important to mention that all modules are intended to provide

correlated and calculated information for cybersecurity experts to analyze and define their own conclusions about the organization under investigation. However, the prioritization process needs further analysis that will be presented in the following section.

**Table 4.** Prioritization table: Extract of 11 vulnerabilities with calculated values shown through the prioritization table in the prototype.

| ip | cve | tr | ep | poe | popI | pqt | po | Impact | rf |
|------|--------------|-----|----|------|------|-----|------|--------|------|
| IP-6 | CVE-2012-2688 | 1   | 1  | 0.25 | 0.73 | 1   | 0.66 | 1      | 0.66 |
| IP-5 | CVE-2016-2842 | 1   | 1  | 0.2  | 0.73 | 1   | 0.64 | 1      | 0.64 |
| IP-5 | CVE-2016-0799 | 1   | 1  | 0.2  | 0.73 | 1   | 0.64 | 1      | 0.64 |
| IP-5 | CVE-2016-0705 | 1   | 1  | 0.2  | 0.73 | 1   | 0.64 | 1      | 0.64 |
| IP-9 | CVE-2016-0799 | 1   | 1  | 0.2  | 0.73 | 1   | 0.64 | 1      | 0.64 |
| IP-9 | CVE-2016-2842 | 1   | 1  | 0.2  | 0.73 | 1   | 0.64 | 1      | 0.64 |
| IP-9 | CVE-2016-0705 | 1   | 1  | 0.2  | 0.73 | 1   | 0.64 | 1      | 0.64 |
| IP-6 | CVE-2011-3268 | 0.8 | 1  | 0.25 | 0.73 | 1   | 0.59 | 1      | 0.59 |
| IP-6 | CVE-2012-2376 | 0.6 | 1  | 0.25 | 0.73 | 1   | 0.53 | 1      | 0.53 |
| IP-6 | CVE-2011-3192 | 1   | 1  | 0.25 | 0.73 | 1   | 0.66 | 0.78   | 0.51 |
| IP-5 | CVE-2016-6304 | 1   | 1  | 0.2  | 0.73 | 1   | 0.64 | 0.78   | 0.5  |

Because the data are correlated to achieve a resulting value of *RiskFactor*, it was possible to validate the value obtained for the illustrative example in Section (Illustrative Example) with the value obtained in the prototype, ensuring the correctness of the calculation process prototype.

*5.2. Prioritization Analysis*

To analyze the prioritization process by calculating the *RiskFactor*, it is essential to know the data resulting from the environment variables since they are the ones that define the prioritization order. According to the point of view of an attacker, the vectors and ways of compromising a network are various and very ingenious. However, all attack processes begin with investigating the information that is available in the wide world of the Internet [35–37,48]. The previous process of knowing how difficult it is to compromise an organization is decisive for an attacker or group of attackers to spend their resources and time. For this reason, all the information that organizations allow to be leaked on the Internet is of vital importance to become a target or not [34,38,39].

Initially, it is essential to know the number of vulnerabilities the organization presents in general. The variable *tv* is responsible for providing us with this information. It is the initial parameter to define whether an automated prioritization process is needed or if the expert in charge has sufficient knowledge.

This variable reflects the time factor since a greater number of vulnerabilities suggests a greater analysis time. If so, our prioritization model applies; otherwise, it is not needed [49,50]. Environment variables can be correlated based on the characteristics and number of vulnerabilities presented by each scanned IP. In Figure 7, the corresponding details are shown. It can be seen that the maximum number of vulnerabilities that an IP contains is 123, and the lowest is 6. These values are essential because they allow for defining whether the number of vulnerabilities establishes the trend of the risk factor; that is, the higher the number of vulnerabilities, the higher the risk factor score.

The vulnerability CVE-2017-9798 presented throughout this study is found in IP-5, IP-7, IP-9, and the prioritization values are 0.23, 0.07, and 0.22, respectively. Consequently, for IP-5, the probability of this vulnerability occurring is higher, giving it a higher priority for attention. If we look at Figure 8, we can see that the existence of the variable *pqt* indicates that the risk factor increases in an IP. Regarding this parame-

ter, we confirm that the model works according to the theoretical approach presented in Section (Probability for Query Tags (PQT)), where it is explained that, when Shodan offers this variable, it shows more information about the place where the vulnerability resides, giving the attacker a more extensive knowledge [20,22,51,52].

In addition, we have the variables of vulnerability. In Figure 9, we can see the trend of the variables *tr* and *ep* that are inversely proportional. We have many vulnerabilities in IPs that maintain a high percentage of referrals. Therefore, the probability of exploitation is adjusted downwards as the correction is final [4,12,18]. In addition, a particular case is observed in IP-2, which presents a total of six vulnerabilities, none of which contains more than ten references. In addition, the CVSS scores on three of the six vulnerabilities are one high and two critical. Despite having the last year in the CVE-ID, an increased risk is still identified, which indicates that vulnerabilities do not constantly adjust downward over the years. This shows that the reference year in the CVE-ID is an unpredictable variable. It will depend on the context of interpretation and the additional information that allows it to be correlated with the probability of exploitation. Therefore, it is concluded that it does not have a fixed trend that indicates lower importance in older vulnerabilities.

According to this study, we can define the IPs' predominant variables. As can be seen in Figure 10, if only CVSS would be used to prioritize vulnerabilities, IP-2 would be the first to be addressed. However, this IP is the lowest priority based on the risk factor calculation. Amankwah et al. [1], Dobrovoljc et al. [3], and Keramati [2] mention that CVSS is usually very generic and does not meet the specificity necessary for an accurate prioritization, which is demonstrated by the trend in Figure 10. Furthermore, if we look at Figure 11, it is clear that CVSS would have a very dysfunctional prioritization order in terms of the organization where the examined vulnerabilities are found.

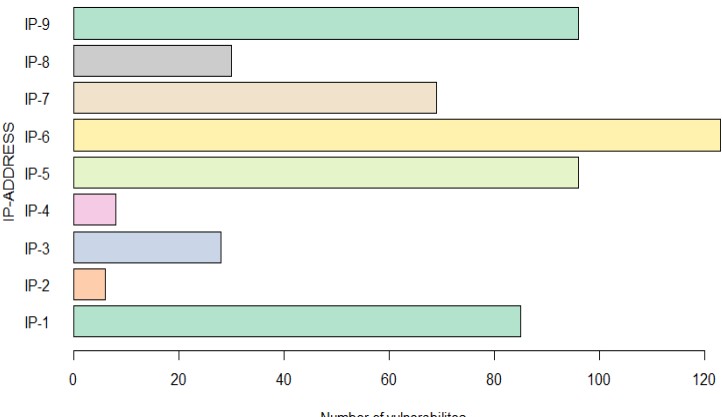

**Figure 7.** Number of Vulnerabilities per IP. A total of 541 vulnerabilities were identified across nine IPs of the same organization.

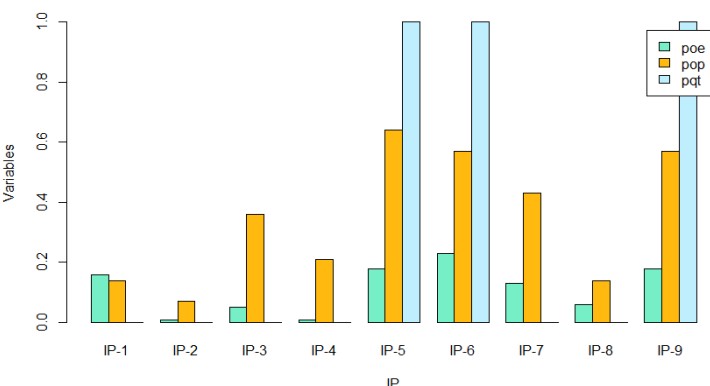

**Figure 8.** Variable IPs. When the pqt variable is present in an IP, the poe and pop variables tend to increase as Shodan reveals more information. The IPs with high levels of sensitive and confidential information are exposed, allowing the collection of a more significant number of environment variables. Therefore, the risk and the probability of occurrence of an event are greater; that is, it is a directly proportional relationship.

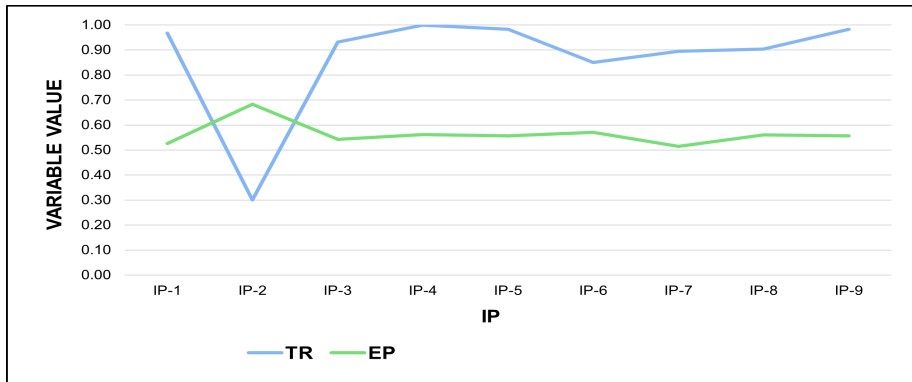

**Figure 9.** Variable vulnerabilities. The relationship between these variables is inversely proportional. When the number of references is low, the vulnerability maintains a high ep since there is a lack of knowledge about the containment and mitigation of the vulnerability.

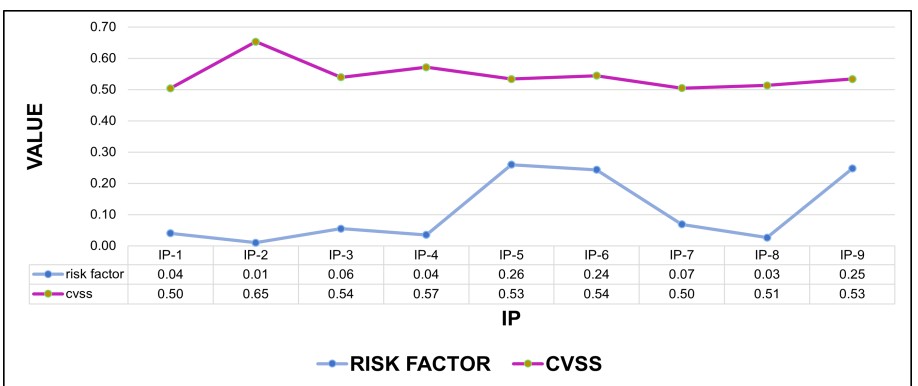

**Figure 10.** CVSS trend and risk factor by IP. The environment variables evaluated and applied for calculating the $rf$ differ from the value assigned by CVSS to a vulnerability since the additional values mean that a vulnerability does not maintain a total impact on the organization. This graph shows that, even with a high CVSS score, the risk factor can be different; i.e., there is no absolute consistency in prioritizing through CVSS.

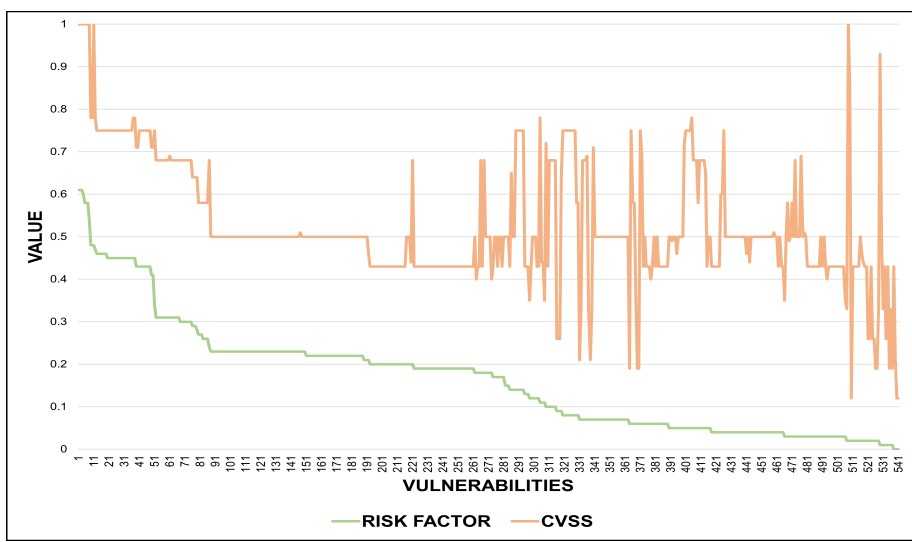

**Figure 11.** CVSS trend and risk factor by vulnerability. If an organization relies only on CVSS, it runs the risk of wasting much time, evaluating vulnerabilities that do not represent a priority risk because environmental variables mitigate the impact and probability of occurrence. This waste of time generates economic losses and increases the organization's risk.

## 6. Discussion

The present prioritization model allows for identifying characteristics of the information-security vulnerabilities and their environment. Case in point, Dondo [5] mentions that systems based on vulnerability attributes help assess the relative potential risk of computer network assets. For their part, Farris et al. [10] analyze the environment variables where vulnerabilities are found and detect vulnerabilities. The different variables contained in the vulnerabilities, such as the year shown in the CVE code and the ports configured in the IPs of the network, are essential when prioritizing since they can indicate a greater probability of exploitation. It is essential to infer risk exposure and the probability of attack; this allows for the classification of vulnerabilities and defining the priority of attention more effectively than CVSS. According to our results, we identify that this approach is fulfilled both for each vulnerability as shown in Figure 11, and for the set of vulnerabilities identified for each IP as shown in Figure 10.

Within the same context, Amankwah et al. [1] mention that CVSS is criticized for its high sensitivity but low specificity for the exploits used and, therefore, the inconsistency in the severity score. For this reason, the vulnerability variables *tr* and *ep* allow us to evaluate the probability of exploitation from these two vectors. The vectors are based on identifying the existence of an exploit for the vulnerability, according to the information that the NIST presents about the CVE. The results show a more centralized approach as shown in Figure 9 and a higher risk precision that allows for prioritizing attention to vulnerabilities.

As previously mentioned, Sharma and Singh [12] propose a hybrid approach derived from the combination of Vulnerability rating and scoring system (VRSS) and CVSS with two temporal metrics. A quantitative score is obtained from qualitative variables for prioritization in this study. The correction level adjusts the severity score downward, leading to less urgency as the correction becomes definitive. Frei [41] mentions that vulnerabilities are exploited within 30 days of their appearance; therefore, it is correct that a vulnerability is adjusted downward. However, it is adjusted downward as there is a definitive correction. However, when this correction is not applied, the risk remains and grows since it also increases the probability of an exploit. For this reason, we use this approach in reverse, achieving a prioritization that assesses an organization's inability to address legacy vulnerabilities.

Likewise, Deb and Roy [16] present a mathematical implementation to identify the status of the different hosts on the network. On the other hand, Hu et al. [17] propose two

algorithms. The first algorithm aims to provide comprehensive predictive information with the threat scenario. The second algorithm quantifies the threat in the first algorithm at the security risk from the host and network levels. Furthermore, the authors mention that CVSS is a good predictor of impact. Based on this network scenario, the environment variables that Shodan provides us about the IP help us to refine the risk factor; as seen in Figure 8, the IP variables mark the prioritization trend since it is the environment where the impact can materialize.

Finally, our model takes CVSS as a reference point plus an additional analysis on the environment variables, which allows us to define a higher quality in the prioritization since the actual environment is analyzed. However, it is not enough to generate an adequate prioritization value that focuses on the technological infrastructure environment variables where vulnerabilities are identified [14,16,53–55]. Thus, the studies that focused only on improving the CVSS model tend to be complex mathematical models that focus on adding variables that define and characterize vulnerability.

## 7. Conclusions and Future Work

In this study, it was possible to define a prioritization model that focused on the attacker's information to compromise an organization through the exploitation of a vulnerability. CVSS is very useful for quantifying environment variables because its metrics allow comparisons with additional information that an organization presents. In addition, it is essential to have an initial risk based on which the criticality of the vulnerabilities can be addressed on their own. When vulnerabilities are not addressed in organizations, they become easy targets for even moderately skilled hackers. Furthermore, if there is overexposure of information on the technological infrastructure, the probability of an event occurring increases.

The risk factor calculated from the information available to an attacker allows for prioritizing vulnerabilities that are visible to any user on the Internet and allows for evaluating the type of information and its sensitivity. Shodan is a powerful search engine with lots of relevant information when using its APIs. The data extraction, treatment, and correlation process become dynamic, achieving continuous monitoring, which allows for evaluating and improving the security of technological infrastructures. Finally, it has been shown that the environment variables indicate that each organization must evaluate the prioritization of vulnerabilities. However, prioritization is best adjusted when a tailored risk factor is calculated for each environment.

As future work, we plan to refine the environment variables presented in this study with historical values. The objective will be to correlate the results of the vulnerabilities that affected the organization and adjust a generic prioritization model that leverages the data to understand its environment through Artificial Intelligence.

**Author Contributions:** Conceptualization J.R., W.F. and P.A. identified the theoretical constructs and the different elements.; methodology, J.R., W.F. and P.A.; software, J.R.; validation, W.F., M.M. and P.A.; formal analysis, W.F.; investigation, J.R., P.A.; resources, W.F.; data curation, P.A.; writing—original draft preparation, J.R., W.F.; writing—review and editing, M.M.; visualization, M.M.; supervision, W.F., M.M; project administration, W.F.; funding acquisition, W.F. All authors have read and agreed to the published version of the manuscript.

**Funding:** This research received no external funding.

**Acknowledgments:** The authors would like to thank the Universidad de las Fuerzas Armadas-ESPE of Sangolquí, Ecuador, for the resources granted to develop the research project entitled: "Design and Implementation of the IT infrastructure and service management system for the ESPE Academic CERT", coded as PIM-03-2020-ESPE-CERT.

**Conflicts of Interest:** The authors declare no conflict of interest.

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
