# Peer review of "An Environment-Specific Prioritization Model for Information-Security Vulnerabilities Based on Risk Factor Analysis"

_electronics, doi:10.3390/electronics11091334_

Round 1

Reviewer 1 Report

This is an interesting study and the authors have collected a unique dataset using cutting edge methodology. The paper is generally well written and structured. However, in my opinion the paper has some shortcomings in regards to some data analyses and text. Below I have provided numerous remarks on the text as it is often vague and long-winded. In several instances I also suggested to cite more relevant and recent literature.

References are not according to the order of citation.

Figure 9, 10 and 11 are not prepared using scientific, tools hence those are to be regenerated.

Author Response

Cover-letter

Response to Reviewers comments

Manuscript ID: electronics-1561454

Title:  An environment-specific prioritization model for information-security vulnerabilities based on risk factor analysis

Status: Mayor Revisions.

Dear editors,

We would like to thank the involved reviewers for their time, effort, and constructive feedback. We believe that the observations and recommendations we received allowed us to significantly improve the quality of our manuscript. We have addressed all the reviewers' comments with the corresponding responsibility point by point.

Our main focus has been to improve the explanation of the proposed research and detail the achieved contributions. In accordance with the established requirements of the scientific community and industry, this research meets the following criteria: 

  • It responds to a real, delicate, and current problem that increases the security level of the organizations that successfully address it. 
  • It involves all phases of the scientific method (i.e., formulation of the problem, analysis of state of the art, statement of the hypothesis, research design, collection, evaluation, analysis, interpretation, and validation of results). Therefore, it meets the standards of scientific and technical quality. 
  • It regards a real need whose solution requires the design of the mathematical prioritization model applying classic probability theory and risk management. Furthermore, the accompanying implementation follows the current software engineering micro-trends and demonstrates the feasibility of our approach. In other words, this work is a tangible and deliverable product that is currently in production in an academic CERT.

In the following, we clarify some additional issues in response to the reviewers' comments and summarize the revisions that have been incorporated in the revised paper. (Note: Changes in the manuscript are tracked with blue colour font).

Respectfully,

Jorge Reyes, 

On behalf of all co-authors

Response to Reviewer Comments

Point 1: “This is an interesting study and the authors have collected a unique dataset using cutting edge methodology. The paper is generally well written and structured. However, in my opinion the paper has some shortcomings in regards to some data analyses and text. Below I have provided numerous remarks on the text as it is often vague and long-winded. In several instances I also suggested to cite more relevant and recent literature.”

Response:

We thank the reviewer for appreciating the content of our paper and providing thoughtful comments. We have performed various revisions according to the reviewer’s comments, noted in the following.

The data analysis has been better detailed, hoping to meet the expectations for the technical content. Bibliographic references have been increased; however, it is important to mention that the first studies regarding vulnerability prioritization provide the basis for the process to follow since the current widely used frameworks are built upon the mathematical definitions that these initial studies introduced. Since 2017, several experimental works have been presented where the authors generally mention that: "The prioritization algorithms, models and mathematical formulas have problems when applied to a real-world environment, due to the multiple objectives and the intrinsic conflicts present in organizations. The different problems have enormous practical significance because they produce a set of solutions based on the importance assigned to each objective." For this reason, the experimentation and analysis of results bring added value by refining the traditional models that have prevailed since the year 2000 but have been proven to be insufficient when the environment is changing. In such cases, the data provide essential information for knowing the environment where the vulnerabilities have been identified and prioritizing them based on a risk factor

Point 2: “Figure 9, 10 and 11 are not prepared using scientific, tools hence those are to be regenerated.”

Response:

The mentioned figures have been regenerated using the R software environment for statistical computing and graphics to provide a clearer and more readable results analysis.

Point 3: “References are not according to the order of citation.”

Response:

We thank the reviewer for this suggestion that improved the readability of our paper. Following the reviewer’s recommendation, we revised the manuscript to list the references in citation order and not alphabetically by author name as before.

Reviewer 2 Report

Authors proposed a mathematical prioritization model to Identify vulnerability in the network. However, the following comments are as follows.  

  1. Authors mentioned in the abstract that "Large amount of Information transmitted over the network makes it difficult to detect vulnerabilities". But authors formulated prioritization models to detect vulnerabilities based on their network environment variables and characteristics which is contrary to the statement specified in the Abstract. This seems to be blunder. Problem Identification should be based on some reasoning to solve the drawback of disadvantage. 
  2. The proposed model is expected to apply on any advanced specific communication networks such VANETs, Smart Grid network, IoT network, WSN etc. which will be useful to carry out further research. Simply theoretical discussion and showing results on virtual basis may not contribute to the research.
  3. Since, each advanced communication network has its own peculiar vulnerabilities, authors need to go through the specific vulnerabilities of the network communication to identify possible threats.

Author Response

Cover-letter

Response to Reviewers comments

Manuscript ID: electronics-1561454

Title:  An environment-specific prioritization model for information-security vulnerabilities based on risk factor analysis

Status: Mayor Revisions.

Dear editors,

We would like to thank the involved reviewers for their time, effort, and constructive feedback. We believe that the observations and recommendations we received allowed us to significantly improve the quality of our manuscript. We have addressed all the reviewers' comments with the corresponding responsibility point by point.

Our main focus has been to improve the explanation of the proposed research and detail the achieved contributions. In accordance with the established requirements of the scientific community and industry, this research meets the following criteria: 

  • It responds to a real, delicate, and current problem that increases the security level of the organizations that successfully address it. 
  • It involves all phases of the scientific method (i.e., formulation of the problem, analysis of state of the art, statement of the hypothesis, research design, collection, evaluation, analysis, interpretation, and validation of results). Therefore, it meets the standards of scientific and technical quality. 
  • It regards a real need whose solution requires the design of the mathematical prioritization model applying classic probability theory and risk management. Furthermore, the accompanying implementation follows the current software engineering micro-trends and demonstrates the feasibility of our approach. In other words, this work is a tangible and deliverable product that is currently in production in an academic CERT.

In the following, we clarify some additional issues in response to the reviewers' comments and summarize the revisions that have been incorporated in the revised paper. (Note: Changes in the manuscript are tracked with blue colour font).

Respectfully,

Jorge Reyes, 

On behalf of all co-authors

Response to Reviewer Comments

Point 1: “Authors mentioned in the abstract that "Large amount of Information transmitted over the network makes it difficult to detect vulnerabilities". But authors formulated prioritization models to detect vulnerabilities based on their network environment variables and characteristics which is contrary to the statement specified in the Abstract. This seems to be blunder. Problem Identification should be based on some reasoning to solve the drawback of disadvantage”.

Response

We thank the reviewer for bringing to our attention this potential misunderstanding. In the abstract, we specifically mention that: "... the large amount of information transmitted over the network makes it difficult to detect vulnerabilities based on their severity and impact." What we try to convey is that the problem lies in prioritizing the detected vulnerabilities and not in the difficulty of detection itself since there are currently several tools that can efficiently perform this task. As such, we think it would be unnecessary to carry out an investigation focusing only on vulnerability detection. As observed in the state of the art, works from the year 2017 and later are no longer focused only on detecting vulnerabilities. Instead, they study and propose appropriate prioritization processes accounting for the large amount of resulting information regarding vulnerabilities that can potentially make organizations waste resources in investigations that are often unnecessary. For this reason, the devised mathematical model focuses on calculating a risk factor that allows prioritizing vulnerabilities to allocate resources on what is riskier for each organization based on the relevant information about the detected vulnerabilities and the respective environment. To sum up, the model we developed is not about detecting vulnerabilities; it is only about prioritizing them.

In order to avoid any confusion, we changed that sentence within the Abstract as follows: “However, the large amount of information transmitted over the network makes it difficult to prioritize the identified vulnerabilities based on their severity and impact.”

Point 2: “The proposed model is expected to apply on any advanced specific communication networks such VANETs, Smart Grid network, IoT network, WSN etc. which will be useful to carry out further research. Simply theoretical discussion and showing results on virtual basis may not contribute to the research”

Response:

We want to thank the reviewer for their constructive comments, as they helped improve the manuscript's quality with this specific observation. In effect, our model can be applied to any communications network (i.e., VANETs, Smart Grid network, IoT network, WSN, etc.) that uses the Shodan platform in its security scheme. Shodan is a search engine that aims to locate vulnerabilities in all types of devices connected to the Internet, from routers, APs, IoT devices to security cameras; this allows technicians to have a list of identified vulnerabilities synchronously or asynchronously, however without prioritization.

The results section and the presentation of the prioritization model specify in detail the followed formulation of values, which is not only theoretical. More precisely, it is detailed how the Shodan's vulnerability-related data can provide us with valuable information without having to know the internal technological infrastructure. Such information was generated despite the difficulty organizations have in presenting data about their vulnerabilities since it is well-known how damaging it can be for an organization to reveal that it was compromised or had faced cyber security problems. Security by obscurity has become the favourite tool of organizations to stay safe. Nevertheless, when such incidents are publicly exposed, they trigger problems at the reputational level and can cause significant financial losses. The specifics about why each variable is used are also explained in the manuscript. To conclude, our prioritization model proposes that the data themselves should be the indicators of the problems that technological infrastructures present.

Point 3: “Since, each advanced communication network has its own peculiar vulnerabilities, authors need to go through the specific vulnerabilities of the network communication to identify possible threats”.

Response:

As mentioned in the answer to the second comment, the examined problem is that it is complicated to create a generic vulnerability prioritization model while attempting to know all of the vulnerabilities. Instead, the particular vulnerabilities are those that are giving us the data and information to quantify the respective risk factors. More precisely, it is necessary to understand that the set of vulnerabilities gives us information and not the underlying network. Also, it is crucial to explain why we use Shodan. We employ this tool because, beyond the data of the vulnerabilities that are the core of this study, it provides relevant information regarding the containing infrastructure, giving us a starting point for interpreting, calculating, and prioritizing the order of attention. Additionally, we leverage the OSINT framework because one of the main objectives of this study is to show to the organizations how cybercriminals see them in the light of these tools, that is, how weak or strong their technological infrastructures appear to attackers. Throughout the study, it has been tried not to violate the privacy of organizations because, as already mentioned, previous experience in cybersecurity cases worldwide tells us that no organization wants to have reputational problems that can be reflected in losses in their financial balances. Taking the above into account, the model proposed in our study defines a risk factor per organization based on the exposure of its technological infrastructure to the virtual world, without the need to know its internal details.

Reviewer 3 Report

Section with overview gives a detailed explanation of existing research in this area. 
Section "research methodology" explains the existing methodologies and gives a short overview of the benefits presented results.

Section "results analyses" in a good way show what has been done in these researches and advantages of the proposed methodology.

Author Response

Cover-letter

Response to Reviewers comments

Manuscript ID: electronics-1561454

Title:  An environment-specific prioritization model for information-security vulnerabilities based on risk factor analysis

Status: Mayor Revisions.

Dear editors,

We would like to thank the involved reviewers for their time, effort, and constructive feedback. We believe that the observations and recommendations we received allowed us to significantly improve the quality of our manuscript. We have addressed all the reviewers' comments with the corresponding responsibility point by point.

Our main focus has been to improve the explanation of the proposed research and detail the achieved contributions. In accordance with the established requirements of the scientific community and industry, this research meets the following criteria: 

  • It responds to a real, delicate, and current problem that increases the security level of the organizations that successfully address it. 
  • It involves all phases of the scientific method (i.e., formulation of the problem, analysis of state of the art, statement of the hypothesis, research design, collection, evaluation, analysis, interpretation, and validation of results). Therefore, it meets the standards of scientific and technical quality. 
  • It regards a real need whose solution requires the design of the mathematical prioritization model applying classic probability theory and risk management. Furthermore, the accompanying implementation follows the current software engineering micro-trends and demonstrates the feasibility of our approach. In other words, this work is a tangible and deliverable product that is currently in production in an academic CERT.

In the following, we clarify some additional issues in response to the reviewers' comments and summarize the revisions that have been incorporated in the revised paper. (Note: Changes in the manuscript are tracked with blue colour font).

Respectfully,

Jorge Reyes, 

On behalf of all co-authors

Round 2

Reviewer 2 Report

All my previous comments are addressed properly